# Hidradenitis Suppurativa and Comorbid Disorder Biomarkers, Druggable Genes, New Drugs and Drug Repurposing—A Molecular Meta-Analysis

**DOI:** 10.3390/pharmaceutics14010044

**Published:** 2021-12-26

**Authors:** Viktor A. Zouboulis, Konstantin C. Zouboulis, Christos C. Zouboulis

**Affiliations:** 1Faculty of Medicine, Universitaetsklinikum Hamburg-Eppendorf (UKE), 20251 Hamburg, Germany; viktor.zouboulis@stud.uke.uni-hamburg.de; 2Department of Chemistry and Applied Biosciences, Swiss Federal Institute of Technology (ETH) Zurich, 8092 Zurich, Switzerland; zouboulk@ethz.ch; 3Departments of Dermatology, Venereology, Allergology and Immunology, Dessau Medical Center, Brandenburg Medical School Theodor Fontane and Faculty of Health Sciences Brandenburg, 06847 Dessau, Germany

**Keywords:** hidradenitis suppurativa, acne inversa, transcriptome, proteome, comorbid disorder, biomarker, drug repurposing, signaling pathway, druggable gene

## Abstract

Chronic inflammation and dysregulated epithelial differentiation, especially of hair follicle keratinocytes, have been suggested as the major pathogenetic pathways of hidradenitis suppurativa/acne inversa (HS). On the other hand, obesity and metabolic syndrome have additionally been considered as an important risk factor. With adalimumab, a drug has already been approved and numerous other compounds are in advanced-stage clinical studies. A systematic review was conducted to detect and corroborate HS pathogenetic mechanisms at the molecular level and identify HS molecular markers. The obtained data were used to confirm studied and off-label administered drugs and to identify additional compounds for drug repurposing. A robust, strongly associated group of HS biomarkers was detected. The triad of HS pathogenesis, namely upregulated inflammation, altered epithelial differentiation and dysregulated metabolism/hormone signaling was confirmed, the molecular association of HS with certain comorbid disorders, such as inflammatory bowel disease, arthritis, type I diabetes mellitus and lipids/atherosclerosis/adipogenesis was verified and common biomarkers were identified. The molecular suitability of compounds in clinical studies was confirmed and 31 potential HS repurposing drugs, among them 10 drugs already launched for other disorders, were detected. This systematic review provides evidence for the importance of molecular studies to advance the knowledge regarding pathogenesis, future treatment and biomarker-supported clinical course follow-up in HS.

## 1. Introduction

Hidradenitis suppurativa/acne inversa (HS) is a chronic, inflammatory, recurrent, debilitating skin disease of the hair follicle that usually presents after puberty with painful, deep-seated, inflamed lesions in the apocrine gland-bearing areas of the body, most commonly at the axillae, inguinal and anogenital regions [1]. A consistent finding, regardless of disease duration, is follicular hyperkeratosis, leading to follicular rupture, inflammation and possible secondary bacterial colonization. The deep part of the follicle appears to be involved. HS is further associated with an initial lymphohistiocytic inflammation, granulomatous reaction, sinus tract formation and scarring [2].

Current own transcriptome and proteome studies highlighted a panel of immune-related drivers in HS, which induce an innate immunity response in epithelial skin cells in a targeted manner [3]. An inflammatory process coupled to impaired barrier function and bacterial activity were detected at the follicular and epidermal keratinocyte and at a minor grade at the skin-gland level. In addition, the adipose tissue was shown to be involved in HS at a real-world immune histochemical study [4].

Despite the beneficial therapeutic effectiveness of several compounds [5,6], treatment of HS is still challenging, since most patients only respond partially with subsequent recurrences. The large unmet need of new therapies requires the elucidation of disease-driving mechanisms and the recognition of the skin compartment initially involved [7,8]. This need can be covered by the development of novel therapeutic regimens for HS [9,10] or by drug repurposing through drug–gene interaction profiling [11,12].

New technology, including inverse virtual screening [13] and computational drug repurposing screening approaches [14], are widely engaged in identifying existing compounds as potential drugs for various diseases. The interaction level of disease and compound molecular profile patterns defines the probability of therapeutic activity of a certain drug. The aim of this study is to provide a wide and robust application of molecular pharmacology in HS through a systematic review of the relevant literature and identification of key molecular mediators in a real-world setting. Using the latter data, therapeutic agents that are currently available or under development for other indications are identified and potential paths for use in the medical management of HS are proposed.

## 2. Materials and Methods

### 2.1. Literature Search

This systematic review was conducted and narrated in accordance with the Preferred Reporting Items for Systematic Reviews and Meta-Analyses (PRISMA) [15] utilizing datasets from publicly available studies, as previously described [11]. A rigorous search of academic databases including PubMed, Web of Science and Ovid databases through August 2021 was conducted. A search strategy predefined and adapted for each aforementioned database included the following keywords: (transcriptome OR proteome OR biomarker(s) OR repurposing OR repositioning OR reprogramming) AND (hidradenitis suppurativa OR acne inversa OR Verneuil’s disease). Additional records were obtained through the Gene Expression Omnibus, National Institutes of Health (Bethesda, MD, USA) [16] and the citation search of the bibliographic records obtained from the academic databases. There were no search filters pertaining to language or publication year.

### 2.2. Study Selection

First the duplicates among bibliographic records were removed. Titles and abstracts were then scrutinized by two reviewers (V.A.Z. and K.C.Z.) working independently according to predefined inclusion and exclusion criteria. This was followed by scrutiny of full texts of eligible studies. Discrepancies were resolved by discussion with the senior investigator (C.C.Z.). After eligible studies were identified, their bibliographies were screened for studies judged suitable for inclusion. Original investigations of HS molecular signatures and protein studies followed by the identification of molecular mediators were selected for further analysis.

### 2.3. Data Extraction

Data pertaining to characteristics of publications under study and quantitative data were extracted by two of the reviewers (V.A.Z. and K.C.Z.) working independently using a predetermined customized extraction form. Characteristics of publications included publication year and affiliation of corresponding authors. Molecular characteristics included transcriptome and/or proteome of HS, and drug repurposing/repositioning/reprogramming.

### 2.4. Data Analysis

Qualitative gene/protein data from the studies were pooled to detect HS signature pathways. Gene nomenclature was verified through the HUGO Gene Nomenclature Committee, European Bioinformatics Institute (Cambridge, UK) public domain [17]. Gene taxonomy was assessed through the biological DataBase network, National Cancer Institute (Frederick, MD, USA) [18]. The molecular pathways were assessed according to the g:Profiler, University of Tartu (Tartu, Estonia) [19], the Kyoto Encyclopedia of Genes and Genomes [KEGG, gene ontology (GO); Kyoto, Japan] [20], the Reactome (REAC), Ontario Institute for Cancer Research (Toronto, ON, Canada), New York University (New York, NY, USA), Oregon Health and Science University (Portland, OR, USA) and the European Molecular Biology Laboratory—European Bioinformatics Institute (Heidelberg, Germany) [21], the WikiPathways (WP) [22] and the Human Phenotype Ontology (HP; The Jackson Laboratory for Genomic Medicine, Farmington, CT, USA) [23] public domains. Random effects were applied throughout the analysis due to expected clinical heterogeneity encountered in different studies supported by g:Profiler [19]. This approach allows heterogeneity in the data to be addressed by considering that differences between studies are random.

### 2.5. Drug Repurposing Sources

For drug repurposing, the detected overall HS molecular signature was compared with the drugs’ molecular signatures of The Drug Repurposing Hub public domain, Eli and Edy L. Broad Institute, MIT and Harvard University (Cambridge, MA, USA) [24] and the Gene Cards, Weizmann Institute of Science (Rehovot, Israel) [25] public domains.

### 2.6. Statistics

Statistics were automatically performed by the applied public domains used [19,20,21,22,23].

## 3. Results

### 3.1. Study Selection Process

A total of 123 bibliographic records were identified after electronic database searches, 36 through other sources and six through bibliographic record citation search. Among them, 61 records were removed as duplicates, leaving 104 titles and abstracts to be screened. After careful screening and manual search, six records were excluded based on title and abstract and 49 records due to inappropriate design and two records due to overlapping data sets with another record, resulting in 47 studies that were included in the quantitative synthesis [3,4,11,26,27,28,29,30,31,32,33,34,35,36,37,38,39,40,41,42,43,44,45,46,47,48,49,50,51,52,53,54,55,56,57,58,59,60,61,62,63,64,65,66,67,68,69] (Figure 1).

### 3.2. Differentially Expressed Genes and Proteins in HS

The comparison of lesional skin vs. non-lesional skin as well as of blood of patients vs. controls at the mRNA and protein levels (cumulatively reported as “targets”) without restrictions revealed 386 differentially expressed genes (DEGs) in HS (Appendix A).

### 3.3. HS Biomarkers

DEGs and differentially expressed proteins in blood and involved skin of HS patients in comparison to controls in at least two relevant articles or two targets were defined as HS biomarkers. Among the 109 detected genes/proteins out of the 386 genes/proteins detected without restrictions, which fulfilled this requirement, 43 DEGs (including the coding genes of detected differentially expressed proteins) have been described in 2/4 targets in two articles, seven in 3/4 targets (*CXCL10*, *IL6*, *IL17A*, *IL36A*, *IL36G*, *S100A8*, *S100A9*) and none in all four targets (Table 1). Additional 10 DEGs have been described in 2/4 targets, however, in a diversified direction (upregulated/downregulated). Among the 109 HS biomarkers, 65 are druggable.

### 3.4. Enrichment Analysis of HS-Associated Genes

The 386 detected HS-associated DEGs and the 109 HS biomarkers were enriched into relevant signaling pathways, which were assessed according to the g:Profiler [19], the KEGG GO, [20], the REAC [21], the WP [22] and the HP [23] public domains in order to identify the major organismal and signal transduction pathways involved in HS. Gene clustering in chromosome 2 and 4 was detected.

Among the 386 HS-associated DEGs, 101 genes were enriched in the cytokine–cytokine (C–C) receptor interaction pathway (−log_10_ = 2.5 × 10^−74^), 51 in the JAK-STAT signaling pathway (2.6 × 10^−34^), 39 in the chemokine signaling pathway (2.7 × 10^−18^), 32 in the IL-17 signaling pathway (1.8 × 10^−22^), 31 in the Th17 cell differentiation pathway (2.6 × 10^−18^), 28 in the Toll-like receptor (TLR) pathway (2.2 × 10^−16^) and 26 in the inflammatory bowel disease pathway (3.6 × 10^−26^) (Appendix A).

Furthermore, 45 HS biomarkers were enriched in the C–C receptor interaction pathway (5.6 × 10^−43^, Figure 2, 19 in the IL-17 signaling pathway (8.8 × 10^−19^, Figure 3), 19 in the JAK-STAT signaling pathway (6.0 × 10^−14^, Figure 4), 18 in the inflammatory bowel disease pathway (1.1 × 10^−20^), 18 in the rheumatoid arthritis pathway (1.2 × 10^−17^), 13 in the Th17 cell differentiation pathway (1.5 × 10^−9^), 13 in the lipid and atherosclerosis pathway (1.2 × 10^−5^), 10 in the TLR pathway (4.3 × 10^−6^), 9 in C-type leptin receptor signaling pathway (6.1 × 10^−5^), 8 in the tumor necrosis factor (TNF) signaling pathway (1.1 × 10^−3^) and 7 in the type I diabetes mellitus pathway (8.5 × 10^−6^) (Figure 5).

Concerning the individual cytokine signaling, IL-17, IL-4, IL-13, IL-10, IL-20 family, IL-1 family, IL-18, IL-36, IL-2 family, IL-21 and IL-12 family signaling included DEGs in HS (Figure 5).

Epithelial differentiation signaling dysregulation in HS was represented by the epidermal growth factor receptor (EGFR), IL-1, IL-1 receptor, formation of the cornified envelope, TLRs and antimicrobial peptides (Figure 5).

Metabolic/obesity-associated dysregulation in HS was detected through type I diabetes mellitus signaling, lipid and atherosclerosis, C-type leptin receptor signaling, estrogen-dependent nuclear events and extranuclear signaling, adipogenesis and resistin signaling (Figure 5).

Interestingly, infection-indicating signaling pathways did not exhibit any major involvement in our study (Figure 5).

At last, the REAC evaluation of globally involved pathways [70] revealed the innate immune system, the cytokine signaling in immune system (major pathways: regulation of *IFNG* signaling), signal transduction (nuclear receptor, *GPCR* and leptin pathways) and developmental biology (formation of the cornified envelope pathway) pathways as the mainly HS-associated ones (Appendix A).

The protein-based connectivity map occurring from an assumed gene biomarker translation (103 proteins our of 109 genes) resulted in 2465 interactions compared with the expected 531 interactions (4.64-fold; *p* < 0.0001), a result that indicates a robust strong protein–protein association in HS (Figure 6). On the other hand, the protein-based connectivity map occurring from the 386 HS-associated DEGs (372 proteins out of 386 genes) resulted in 19,823 interactions compared with the expected 6502 interactions (3.05-fold; *p* < 0.0001), indicating that the biomarker selection procedure increased the HS/protein association.

### 3.5. Enrichment Analysis of HS Druggable Genes

Among the 386 HS-associated DEGs, 105 druggable genes were recognized. With the 11 additional druggable genes described by Zouboulis et al. [12], namely *ABAT*, *ADRA1A*, *CYP3A4*, *GRM4*, *HRH1*, *OPRD1*, *OPRM*, *PRKAB1*, *PTGS1*, *PTGS2* and *SLC6A4*, the overall detected druggable genes in HS are 116.

The 116 druggable genes were enriched in relevant signaling pathways according to the KEGG GO [20] and the Gene Cards [25] public domains to identify the major targeted organismal and signal transduction pathways (Appendix A). Twenty-two druggable genes were enriched in the lipid and atherosclerosis pathway (8.4 × 10^−13^), 19 in the JAK-STAT signaling pathway (6.2 × 10^−12^), 17 in the Th17 cell differentiation pathway (5.2 × 10^−13^), 17 in the IL-17 signaling pathway (6.0 × 10^−14^), 16 in the inflammatory bowel disease pathway (1.5 × 10^−16^), 14 in the TLR signaling pathway (6.0 × 10^−14^), 14 in the C-type leptin receptor signaling pathway (2.4 × 10^−9^) and 13 in the TNF signaling pathway (8.4 × 10^−8^).

### 3.6. Study Drugs and Drug Repurposing for HS

The majority of registered, studied or off-label administered drugs modify HS-associated DEGs. On the other hand, the evaluation of the detected 105 HS-associated druggable genes proposed 452 potentially therapeutic compounds, among them 120 launched drugs, 178 compounds in clinical studies and 154 in preclinical evaluation (Appendix A). Among these potentially therapeutic compounds, the 31 drugs, which regulate three or more genes with all of them being HS-associated DEGs or at least four genes with 60% of them been DEGs were classified as probable repurposing drugs for HS (Table 2).

## 4. Discussion

### 4.1. HS Pathogenesis

Inflammation doubtlessly plays a major role in the pathogenesis of HS [3,7,8]. Proteome studies provide evidence that the innate immunity system and both *IL-1* and *IL-17* signaling pathways are activated in HS lesions and circulating neutrophils [27,40,45,71,72,73], findings that have been confirmed in our systematic review. In addition, Th17 differentiation of CD4+ lymphocytes is activated in HS [57]. Among others, Kelly et al. [38] provided evidence that CD45+CD4+ T cells are responsible for IL-17 production and CD11c+CD1a-CD14+ dendritic cells are the main producers of IL-1β in lesional HS skin. The IL-17 cytokine family has been linked to the pathogenesis of diverse autoimmune and inflammatory diseases and also plays an essential role in host defense against extracellular microorganisms [2,74]. IL-17 has been shown to increase the expression of skin antimicrobial peptides, including human β-defensin 2, psoriasin (S100A7) and calprotectin (S100A8/9) in keratinocytes and of a number of cytokines attracting neutrophils [75]. Thus, IL-17 may contribute to inflammation by increasing the influx of neutrophils, dendritic cells and memory T cells into the lesions. On the other hand, the involvement of *IL-1* signaling pathway is also prominent in HS with upregulation of molecules causing immune cell infiltration and extracellular matrix degradation and could be reversed by application of IL-1 receptor antagonist [40,76]. *IL1B* signaling pathway-associated genes, such as *IL1R1*, *IL1RN*, *IFNG*, *IL6*, *IL18*, *IL18R1*, *IL32*, *IL33*, *IL36A*, *IL36B*, *IL36G*, *IL36RN*, *IL37*, *TLR2*, *TLR3*, *TLR4*, *S100A7*, *S100A7A*, *S100A8*, *S100A9* and *S100A12* were HS-associated DEGs, as detected in our systemic review. 

The inflammatory process in HS seems to be coupled with impaired barrier function, altered epidermal cell differentiation, formation of the cornified envelope, TLRs and antimicrobial peptides [3], the latter not being associated with any infection, as clearly shown in the present study. These events have been observed at the follicular and epidermal keratinocytes and at a minor grade at the skin glands [3]. Moreover, we could confirm a dysregulated expression pattern of serpins, small proline-rich proteins and certain keratins, which further support the involvement of the follicular infundibulum in the initiation of the lesions, especially at the anatomic area of communication with the apocrine gland duct and the ductus seboglandularis [3].

Although HS has well-documented associations with the metabolic syndrome, which is characterized by systemic inflammation identified at a molecular level [77], the role of adipose tissue in HS has barely been investigated. Obesity is currently shown to represent the primary risk factor in HS at the molecular level [4,28]. A chronic low-grade subclinical inflammatory response is strongly implicated in the pathogenesis of insulin resistance and metabolic syndrome. The clinically relevant peroxisome proliferator-activated receptor (PPAR) pathway was down-regulated in adipocytes of HS lesions [4]. In agreement with these data, reduced serum levels of adiponectin were currently found in non-diabetic patients with HS [28]. Since adiponectin inhibits the production of TNF-α, IL-6 and chemokines of human macrophages the upregulation of *ADIPOQ* and *PLIN1*, shown in this systematic review, might be beneficial in HS treatment. Indeed, thiazolidine derivatives act as PPARγ agonists and effectively increase the adiponectin concentration and adipogenic gene expression [28,78]. Unsaturated fatty acids, eicosanoids and non-steroidal anti-inflammatory drugs function in a similar manner [79]. Further metabolic pathways, e.g., the IGF transport and uptake of IGF-binding proteins pathway, type I diabetes mellitus signaling, lipid and atherosclerosis, C-type leptin receptor signaling, estrogen-dependent nuclear events and extranuclear signaling and RETN signaling, encoding resistin, are dysregulated in HS, as shown in the present review.

In conclusion, inflammatory signaling, mainly innate immunity signaling pathways, mostly that of IL-1 and IL-17, epithelial differentiation signaling pathways, primarily of follicular keratinocytes and skin gland duct cells and metabolic signaling pathways, especially that of obesity/adipogenesis, represent pathogenetic HS cascades, whose activity may be targeted by future therapeutic means.

### 4.2. HS Comorbid Disorders

HS has been associated with a variety of comorbid disorders, such as inflammatory bowel diseases, especially Crohn’s disease, axial spondylarthritis without or with follicular occlusion, triad signs, genetic keratin disorders associated with follicular occlusion, such as pachyonychia congenita, steatocystoma multiplex, Dowling-Degos disease without and with arthritis, as well as other genetic disorders, such as keratitis–ichthyosis–deafness syndrome and Down syndrome [80]. Moreover, HS has been associated with reduced quality of life, metabolic syndrome, sexual dysfunction, working disability, depression and anxiety. Like in psoriasis, HS patients have higher prevalence of cardiovascular disease risk factors and suicide risk [81]. At last, the development of epithelial tumors on chronic HS lesions at the anogenital region may be considered as the consequence of chronic severe inflammatory skin disease. The current work has provided molecular evidence of HS association with inflammatory bowel disease pathway, rheumatoid arthritis pathway, type I diabetes mellitus signaling, lipid and atherosclerosis and adipogenesis signaling.

### 4.3. Study Drugs and Drug Repurposing for HS

In addition to the only registered drug in HS, namely adalimumab [9,82,83], the majority of studied and off-label administered drugs also regulate differentially expressed genes and their proteins in HS, as shown in the present review [10,65,76,81,82,83,84,85,86,87,88,89,90,91,92,93,94,95]. On the other hand, the 452 HS-associated druggable genes proposed can mostly be classified in receptor ligands, enzyme/protein inhibitors, JAK-STAT inhibitors, PI3K inhibitors, sodium/potassium/calcium channel activators and MMP inhibitors. Additionally, Gentamicin, Ibudilast, Spironolactone, Trastuzumab, Thalidomide, Apremilast, Glucosamine, Interferon-a-2b, Binimetinib and Midostaurin have previously been reported as repurposing drugs for HS [11]. The majority of the 31 probable repurposing drugs shown in Table 2 are JAK inhibitors, with cytokine inhibitors, such as anti-IL-17 compounds, tyrosine kinase receptor inhibitors, TNF inhibitors, cyclooxygenase inhibitors, EGF receptor inhibitors, MMP inhibitors and PPARγ ligands—among others—being represented. Ten of these drugs, which have not yet been administered in HS, are already launched for other indications and 17 are in clinical studies, not including HS.

## 5. Conclusions

The current review provides robust molecular evidence on the pathogenetic triads of HS, namely upregulated inflammation, dysregulated epithelial cell differentiation and obesity signaling/hormone involvement. In addition, evidence of the negligible role of infectious agents is included. Moreover, HS biomarkers with strong protein–protein connectivity in HS are presented. While adalimumab, the only currently registered drug in HS, and the majority of studied and off-label administered drugs regulate DEGs and their proteins in HS, numerous compounds are eligible for HS repurposing due to their molecular signaling. Among them, 31 compounds are designated probable, following our classification, with 10 of them already being launched for other indications.

## Figures and Tables

**Figure 1 pharmaceutics-14-00044-f001:**
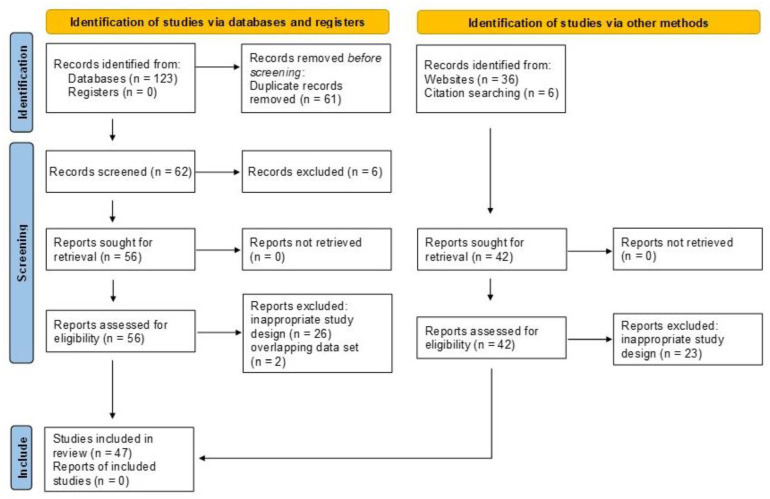
Preferred reporting items for systematic reviews and meta-analyses (PRISMA 2020 [15]) flow diagram.

**Figure 2 pharmaceutics-14-00044-f002:**
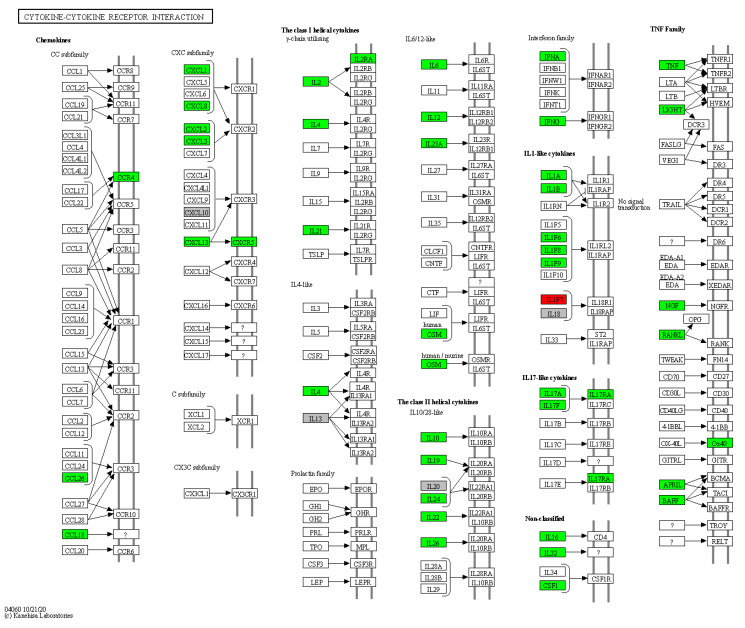
Hierarchical clustering of HS biomarkers in the KEGG GO C-C receptor interaction pathway. Genes which are positively regulated in HS are shown in green color, those downregulated with red color. Gray color corresponds to genes with a diversified reported regulation.

**Figure 3 pharmaceutics-14-00044-f003:**
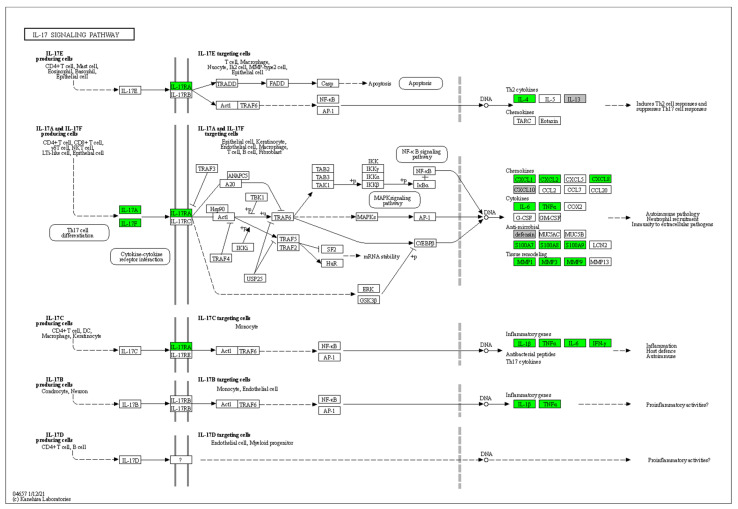
Hierarchical clustering of HS biomarkers in the KEGG GO IL-17 signaling pathway. Genes which are positively regulated in HS are shown in green color. Gray color corresponds to genes with a diversified reported regulation.

**Figure 4 pharmaceutics-14-00044-f004:**
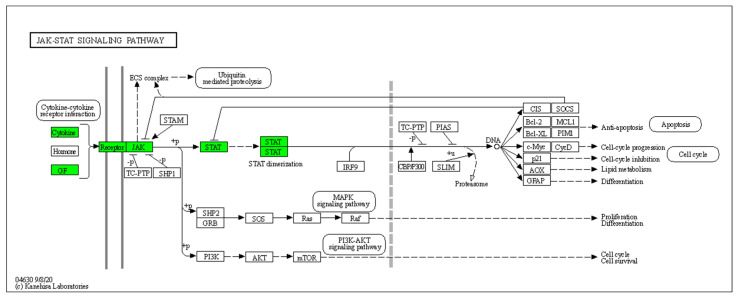
Hierarchical clustering of HS biomarkers in the KEGG GO JAK-STAT signaling pathway. Genes which are positively regulated in HS are shown in green color. Gray color corresponds to genes with a diversified reported regulation.

**Figure 5 pharmaceutics-14-00044-f005:**
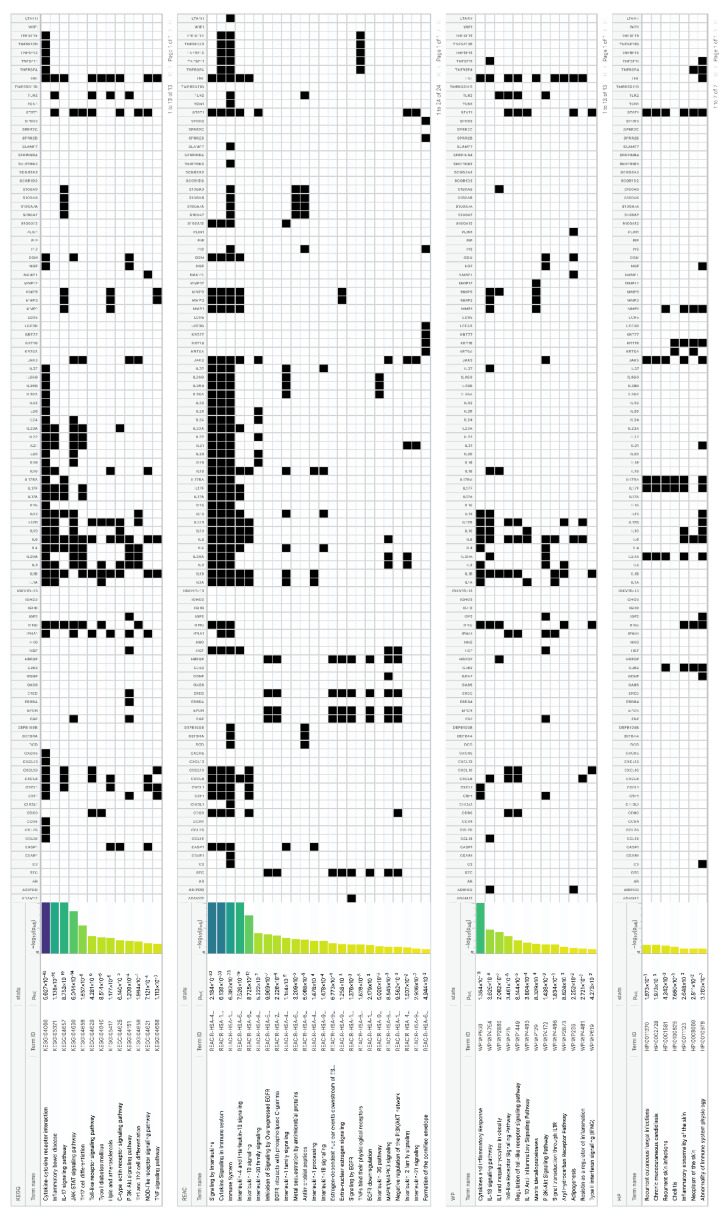
Enrichment of HS biomarkers resulting from the comparison of transcriptomic profiles and protein expression studies between lesional HS and non-lesional skin biopsies and blood samples from HS patients and healthy controls, respectively, in signaling pathways.

**Figure 6 pharmaceutics-14-00044-f006:**
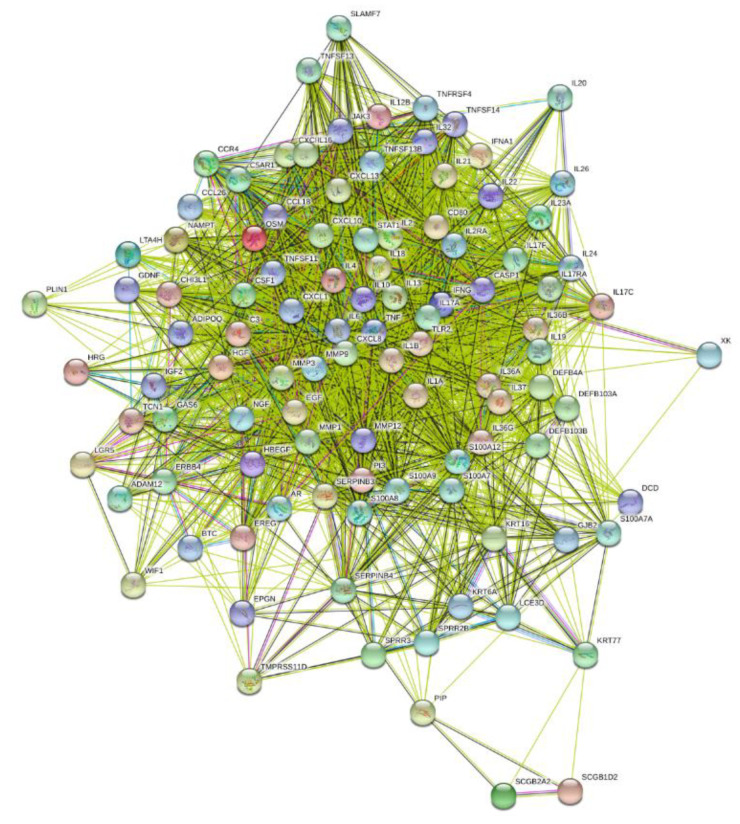
Biomarker-resulting protein-based connectivity map of HS.

**Table 1 pharmaceutics-14-00044-t001:** HS biomarkers resulting from the DEGs after transcriptomic profiling and protein expression studies between lesional HS and non-lesional skin biopsies and blood samples from HS patients and healthy controls, respectively and reported in at least two relevant articles. Bold letters indicate druggable genes. Background: white = similar results reported in one target (biological material) in at least two independent studies; orange = similar results reported in two targets in at least two independent studies; yellow = similar results reported in three targets in at least two independent studies. Gray = diversified result reported in at least two independent studies; + = upregulation; − = downregulation; +/− = diversified dysregulation in different studies; () = lower level of evidence.

	Blood			Skin						
Gene	+/−	mRNA	Protein	+/−	mRNA	Protein	Name	Other skin disorders	HS comorbid disorders	Drugs
ADAM12				+	[3,27]		ADAM Metallopeptidase Domain 12		Down syndrome	
**ADIPOQ**	−		[28]	−	[27]		Adiponectin		Glucose intolerance, metabolic syndrome	Piogitazone
**AR**				+	[3,33,34]	[35]	Androgen receptor	Polycystic ovary syndrome, alopecia	Androgen insensitivity syndrome	Cyproterone acetate, Flutamide, Nilutamide, Bicalutamide, 17α-Propionate, AZD3514
**BTK**				+/(−)	[3,27,33,34]		Betacellulin	Squamous cell carcinoma		Cetuximab
**C3**	−		[27]	+	[30]		Complement C3			Zinc, Zinc acetate
**C5AR1**				+	[3,30]		Complement C5a Receptor 1	Hypersensitivity reaction type III disease		Compstatin, PMX 205, PMX 53, W 54011
**CASP1**				+		[38,39]	Caspase 1	Schnitzler syndrome	Familial Mediterranean fever	Minocyclin
CCL18				+	[27,30]	[43]	C-C Motif Chemokine Ligand 18	Eczema		
CCL26	+		[41]	+	[30]		C-C Motif Chemokine Ligand 26			
CCR4	+	[45]		+	[30,45]		C-C Motif Chemokine Receptor 4	Mycosis fungoides, cutaneous T cell lymphoma, allergic contact dermatitis		
**CD80**				+	[30,38]		CD80 Molecule			Abatacept, Belatacept
CHI3L1	+		[49]	+		[50]	Chitinase 3-Like 1	Erysipelas		
CSF1				+	[3,33,34,40]		Colony-Stimulating Factor 1		Rheumatoid arthritis	
**CXCL1**				+	[27,40,42,44,45]	[40]	C-X-C Motif Chemokine Ligand 1	Kaposi sarcoma		Formic acid
**CXCL8**				+	[30,42,44]	[41]	C-X-C Motif Chemokine Ligand 8	Melanoma		Simvastatin
**CXCL10**	−		[41]	+	[30]	[41]	C-X-C Motif Chemokine Ligand 10			Eldelumab
CXCL13				+	[30,42,45]	[26]	C-X-C Motif Chemokine Ligand 13	T cell lymphoma		
CXCR5				+	[30]	[26]	C-X-C Motif Chemokine Receptor 5	T cell lymphoma		
**DCD**				−	[27,32,33]	[32]	Dermcidin	Netherton syndrome, tinea pedis		Basiliximab, Zinc sulfate
DEFB4A				+/(−)	[3,27,30,32,39,44,45,46]	[3,53]	Defensin β 4A	Tinea corporis, oral candidiasis		
DEFB103B				+	[46,52]		Defensin β 103B			
**EGF**				+	[3,33,34]		Epidermal Growth Factor			Cetuximab, AG 490, CGP 52411, Genistein, Zanubrutinib (receptor antagonist)
EPGN				+	[3,33,34]		Epithelial Mitogen	Seborrheic dermatitis		
**ERBB4**				−	[27,32]		Erb-B2 Receptor Tyrosine Kinase 4			Gefitinib, Afatinib, Fostamatinib, AG 490, CGP 52411, Genistein
EREG				+	[3,33,34]		Epiregulin			
GAS6				+/(−)	[3,33,34]		Growth Arrest Specific 6	Lupus erythematosus		
**GDNF**				+	[3,33,34]	[36]	Glial Cell Derived Neurotrophic Factor			Chondroitin sulphate
**GJB2**				+	[3]	[3]	Gap Junction Protein β2	Keratitis-Ichthyosis-Deafness Syndrome		Carbenoxolone disodium
HBEGF				+	[3,33,34]		Heparin Binding EGF-Like Growth Factor			
**HGF**				+	[3,33,34]		Hepatocyte Growth Factor			Dexamethasone, Neratinib, Erlotinib
**HRG**				+	[3,33,34]		Histidine-Rich Glycoprotein			Zinc sulfate
IFNA1				+	[3,26,30,33,34]		Interferon α1	Cryoblobulinemia		
**IFNG**				+	[3,26,30,33,34,40,44,45,46]		Interferon γ			Oksalazine, Emapalumab, Glucosamine
IGF2				+	[3,33,34]		Insulin-Like Growth Factor 2			
IGHD				+	[27,30]		Immunoglobulin Heavy Constant δ			
IGHG3				+	[27,30]		Immunoglobulin Heavy Constant γ3 (G3m Marker)			
IGKV1D-13				+	[27,30]		Immunoglobulin κ Variable 1D-13			
IGLV				+	[27,30]		Immunoglobulin λ Variable Cluster			
**IL1A**				+	[3,26,30,33,34,40]	[39]	Interleukin 1α	Acne, Irritant dermatitis	Arthritis	Anakinra, Rinolacept, Olanzapine, Pirfenidone, Thalidomide, AMG-108
**IL1B**				+	[26,30,38,40,42,46]	[38,56]	Interleukin 1β	Gingivitis, Muckle–Wells syndrome, Toxic shock syndrome		Canakizumab, Anakinra (receptor antagonist), Rinolacept (receptor antagonist), Minocycline
**IL2**				+	[26,30]		Interleukin 2	Graft-versus-host disease, Leprosy		Suplatast tosylate, Daclizumab (receptor antagonist), Basiliximab (receptor antagonist), Rituxomab, Thalidomide, Cafazolin
**IL2RA**	+		[49,56]	+	[30]		Interleukin 2 Receptor Subunit α		Type 1 diabetes mellitus, Juvenile arthritis	Daclizumab, Basiliximab, Pirfenidone, Thalidomide
**IL4**				+	[3,30,33,34,40]		Interleukin 4	Atopy, Allergic rhinitis, Food allergy		Dupilumab (receptor antagonist), Calcitriol
**IL6**	+		[40]	+	[3,26,30,33,34,40,42]	[40,58]	Interleukin 6			Siltuximab, Tocilizumab (receptor antagonist), Sarilumab (receptor antagonist), Satralizumab (receptor antagonist), Vitamin C, Vitamin E
**IL10**				+	[30,38,44,46]	[52,56]	Interleukin 10			Nicotinamide, Niacin, Cyclosporine A, Methotrexate, Mycofenolate mofetil
**IL12A**				+	[59]	[41]	Interleukin 12A	Adamantiades–Behçet’s disease	Primary biliary cholangiitis	Mycophenolate mofetil, Ustekinumab (IL-12/23), Briakinumab (IL-12/23)
**IL12B**				+	[30]	[36]	Interleukin 12B	Psoriasis		Ustekinumab (IL-12/23), Briakinumab (IL-12/23)
**IL13**				+/(−)	[3,30,45]		Interleukin 13	Allergic rhinitis, Penicillin allergy		Suplatast tosylate, Montelukast, Omalizumab
IL16				+	[30]	[41]	Interleukin 16		Allergic asthma	
**IL17A**	+		[59]	+	[3,30,33,34,38,39,40,42,44,46,60]	[4,36,38,39,41]	Interleukin 17A	Allergic contact dermatitis	Arthritis	Secukizumab, Ixekizumab, Bimekizumab (IL-17A/F), Brodalumab (receptor antagonist), Vidofludimus
**IL17F**				+	[30,39,40,42,45]		Interleukin 17F	Candidiasis, Acute generalized exanthematous pustulosis, Mail diseases		Bimekizumab (IL-17A/F), Brodaluman (receptor antagonist)
**IL17R**				+	[3]	[4]	Interleukin 17 Receptor	Candidiasis	Arthritis	Brodalumab
**IL18**				+/−	[26,30]	[38]	Interleukin 18			IAP antagonist, Iboctadekin + Doxil
IL19				+	[3,30,40]		Interleukin 19	Psoriasis	Inflammatory bowel disease, Arthritis	
IL20				+/−	[30,46]	[46]	Interleukin 20	Psoriasis		
IL21				+	[30,39]		Interleukin 21		Dacryoadenitis, Inflammatory boel disease	
IL22				+/(−)	[3,30,40,42,46]	[46]	Interleukin 22	Candidiasis	Inflammatory bowel disease	
IL22RA1				−	[30]	[46]	Interleukin 22 Receptor Subunit α1		Spondyloarthropathy, rheumatoid arthritis, autoimmune uveitis	
**IL23A**				+	[30,40,61]		Interleukin 23 Subunit α	Autoimmune disease	Inflammatory bowel disease, Arthritis	Guselkumab, Risankinumab, Tildrakizumab, Ustekinumab (IL-12/23), Briakinumab (IL-12/23)
IL24				+	[30,42,46]		Interleukin 24	Melanoma, chronic spontaneous urticaria, psoriasis	Spondylarthropathy	
IL26				+	[42,46]		Interleukin 26	Psoriasis	Inflammatory bowel disease, Crohn’s disease	
IL32				+	[30,40,61]		Interleukin 32	Cutaneous diphtheria		
**IL36A**	+		[62]	+	[30,40,42,45,61]	[39,61]	Interleukin 36α	Psoriasis		Spesolimab (receptor antagonist)
**IL36B**	+		[62]	+		[61]	Interleukin 36β		Periostitis	Spesolimab (receptor antagonist)
**IL36G**	+		[62]	+	[30,40,42,45]	[61]	Interleukin 36γ	Acute generalized exanthematous pustulosis, Psoriasis		Spesolimab (receptor antagonist)
**IL37**				−	[32,33,42]		Interleukin 37	Still’s disease	Inflammatory bowel disease	Ustekinumab (IL-12/23)
**JAK3**				+	[3,30]		Janus Kinase 3		NK cell enteropathy	Decernatinib, Tofacitinib (JAK1/3), Ruxolitinib (JAK1/3), PF-06651600, AT-501, ATI-502, Cerdulatinib (JAK1/2/3, SYK), Delgocitinib (JAK1/2/3), Peficitinib (JAK1/2/3), Zanubrutinib (JAK3/ITR/EGFR), Cercosporamide JAK3/Mnk2)
**KRT6A**				+	[3,32]	[3]	Keratin 6A	Pachyonychia congenita, Lingua plicata, Cheilitis		Zinc, Zinc acetate
KRT16				+	[3,27,30,32]	[3]	Keratin 16	Pachyonychia congenita, palmoplantar keratoderma		
KRT77				−	[27,32,33]	[32]	Keratin 77	Epidermolytic palmoplantar keratoderma, Buschke-Ollendorff syndrome		
LCE3D				+	[32]	[32]	Late Cornified Envelope 3D	Psoriasis		
LGR5				−	[27,32]		Leucine Rich Repeat Containing G Protein-Coupled Receptor 5		Type II diabetes mellitus	
**LTA4H**	−		[27,65]	+	[31]		Leukotriene A4 Hydrolase			Captopril, Dexamethasone, Montelukast
**MMP1**				+	[3,30]	[3]	Matrix Metallopeptidase 1	Epidermolysis bullosa atrophica, Scleroderma		Zinc, Collagenase
**MMP3**				+	[40]	[40]	Matrix Metallopeptidase 3		Coronary heart disease, Arthritis	Pravastatin, Simvastatin, Prothalidone, Lisinopril
**MMP9**				+	[3,30,40]	[3]	Matrix Metallopeptidase 9			Minocycline, Capropril, Simvastatin, Zinc, Zinc acetate
**MMP12**				+	[27,30]		Matrix Metallopeptidase 12	Dermatitis herpetiformis, Middermal elastolysis	Arthritis	Acetohydroxamic acid, Batimastat
**NAMPT**	+		[28,63]				Nicotinamide Phosphoribosyl transferase	Skin aging, pellagra, diabetes mellitus type 2, polycystic ovary syndrome		Nicotinamide, Niacin
**NGF**				+	[3,33,34]	[36]	Nerve Growth Factor			Clenbuterol
OSM				+	[3,26]	[36]	Oncostatin M	Kaposi sarcoma		
PI3				+	[3,27,32,33]	[3]	Peptidase Inhibitor 3	Pustular psoriasis, impetigo herpetiformis, erysipelas		
PIP				−	[27,32]		Prolactin Induced Protein			
**PLIN1**				+/−	[27,48]		Perilipin 1			Rosiglitazone
**S100A7**				+	[3,30,33,39,42,44,46]	[32]	S100 Calcium-Binding Protein A7	Psoriasis, Squamous cell carcinoma	Anal fistula	Ibuprofen, Dexibuprofen, Zinc, Zinc acetate, Zinc chloride
S100A7A				+	[3,27,32]	[3,32]	S100 Calcium-Binding Protein A7A	Psoriasis		
**S100A8**	+		[57]	+	[3,33,34,44]	[3,32]	S100 Calcium-Binding Protein A8			Zinc, Zinc acetate, Zinc chloride, Copper
**S100A9**	+		[57]	+	[3,27,32,33,42,44,46]	[3,32]	S100 Calcium-Binding Protein A9		Crohn’s disease, Rheumatoid arthritis	Zinc, Zinc acetate, Zinc chloride, Calcium
**S100A12**				+	[3,30,32,42]	[3,41]	S100 Calcium-Binding Protein A12	Kawasaki disease	Psoriatic arthritis	Amlexanox, Olopatadine
SCGB1D2				−	[27,32]		Secretoglobin Family 1D Member 2			
SCGB2A2				−	[27,32,33]		Secretoglobin Family 2A Member 2			
**SERPINB3**				+	[3,27,30]	[3]	Serpin Family B Member 3	Squamous cell caecinoma		Phosphoserine
SERPINB4				+	[3,27,30]	[3]	Serpin Family B Member 4	Squamous cell carcinoma		
**SLAMF7**				+	[3,27]		SLAM Family Member 7	IgG4-related disease		Elotuzumab
SPRR2B				+	[32]	[32]	Small Proline Rich Protein 2B	Photosensitive trichothio-dystrophy 1, Autosomal reces-sive congenital ichthyosis		
SPRR2C (pseudogene)				+	[32]	[32]	Small Proline Rich Protein 2C (Pseudogene)			
SPRR3				+	[3]	[3]	Small Proline Rich Protein 3	Genodermatoses		
**STAT1**				+	[3,26,30,44]	[36]	Signal Transducer and Activator of Transcription 1			Methimazole, Niclosamide, Nifuroxazide, Sulforaphane
**TCN1**				+	[3,27,45]	[3]	Transcobalamin 1			Hydroxycobalamin, Cyanocobalamin, Cobalt
**TLR2**				+	[3,68]		Toll-Like Receptor 2	Leprosy, Borreliosis	Colorectal cancer	Adapalene, Cyproterone acetate
**TLR4**				+/−	[26]	[53]	Toll-like Receptor 4			Paclitaxel, Tacrolimus, Cyclobenzaprine
TMPRSS1D				+	[3]	[3]	Transmembrane Serine Protease 11D			
**TNF**				+	[3,26,30,32,33,38,40]	[56]	Tumor Necrosis Factor	Psoriasis, Toxic shock syndrome	Inflammatory bowel diseases, Arthritis	Adalimumab, Infliximab, Golimumab, Etanercept (receptor antagonist), Certolizumab pegol, Thalidomide, Lenalidomide, Pomalidomide, Calcitriol, Bay 11-7821, (R)-DOI, Cannabidiol
**TNFRSF4**	+	[45]		+	[45]		TNF Receptor Superfamily Member 4	Kaposi sarcoma, Graft-versus-host disease, Drug reaction with eosinophilia		OX-40 ligand
**TNFSF11**				+	[30]	[36]	TNF Superfamily Member 11			Letrozole, Thiocolchicoside
**TNFSF13 (APRIL)**				+	[30]	[26]	TNF Superfamily Member 13	Autoimmune diseases	Rheumatoid arthritis	Pomalidomide, TACI-IG
**TNFSF13B (BAFF)**				+	[30]	[26]	TNF Superfamily Member 13b	Autoimmune diseases, Sialadenitis, Sjogren syndrome		Belimumab, Blisibimod, LY2127399, TACI-IG
TNFSF14				+	[30]	[36]	TNF Superfamily Member 14	Herpes simplex	Rheumatoid arthritis	
TNIP1				+/−	[26,30]		TNFAIP3 Interacting Protein 1	Systemic lupus erythematosus, Psoriatic arthritis	Rheumatoid arthritis, Arthritis	
WIF1				−	[27,32]		WNT Inhibitory Factor 1			

**Table 2 pharmaceutics-14-00044-t002:** Probable HS repurposing drugs * and molecular profile of drugs registered ** or off-label administered in HS.

Compound	Function	Gene Regulation	Development Phase
*Probable repurposing HS drugs*			
3,3’-Diindolylmethane	CHK inhibitor, cytochrome P450 activator, indoleamine 2,3-dioxygenase inhibitor	**AR**, HIF1A, **IFNG**, **PI3**	3
AG-490	EGFR inhibitor, JAK inhibitor	**EGFR**, **JAK2**, **JAK3**	preclinical
Andrographolide	tumor necrosis factor production inhibitor	**IL1B**, **IL6**, **NFKB1**, NFKB2, **TNF**	2
Apratastat	matrix metalloprotease inhibitor, tumor necrosis factor production inhibitor	ADAM17, **MMP1**, **MMP13**, **MMP9**	2
Atractylenolide-I	JAK inhibitor	**JAK1**, **JAK2**, **JAK3**	preclinical
AZD1480	JAK inhibitor	**JAK1**, **JAK2**, **JAK3**	1
Balsalazide	cyclooxygenase inhibitor	**ALOX5**, **PPARG**, **PTGS1**, **PTGS2**	launched
BMS-911543	JAK inhibitor	**JAK1**, **JAK2**, **JAK3**	1/2
Ciglitazone	PPARγ agonist	**GPD1**, **PPARG**, **TBXA2R**	2
Curcumol	JAK inhibitor	**JAK1**, **JAK2**, **JAK3**	1
Cyt387	JAK inhibitor	**JAK1**, **JAK2**, **JAK3**	3
Delgocitinib	JAK inhibitor	**JAK1**, **JAK2**, **JAK3**	2
Fedratinib	FLT3 inhibitor, JAK inhibitor	**BRD4**, **JAK1**, **JAK2**, **JAK3**, **TYK2**	launched
Filgotinib	JAK inhibitor	**JAK1**, **JAK2**, **JAK3**, **TYK2**	3
Ganoderic-acid-a	JAK inhibitor	**JAK1**, **JAK2**, **JAK3**	preclinical
JTE-607	cytokine production inhibitor	**IL10**, **IL1B**, **IL6**, **TNF**	2
Latamoxef	Cephalosporine	**DACB**, **MRCA**, **MRCB**, **PBPC**	launched
LXR-623	Liver X receptor agonist	**AR**, **NR1H2**, **NR1H3**, NR1I2, NR3C1	1
NS-018	JAK inhibitor	**JAK1**, **JAK2**, **JAK3**, **TYK2**	1/2
Pacritinib	FLT3 inhibitor, JAK inhibitor	FLT3, **JAK1**, **JAK2**, **JAK3**	3
Paracetamol	cyclooxygenase inhibitor	FAAH, **PTGS1**, **PTGS2**, **TRPV1**	launched
Peficitinib	JAK inhibitor	**JAK1**, **JAK2**, **JAK3**	launched
PF-06651600	JAK inhibitor	**JAK1**, **JAK2**, **JAK3**	2/3
Plerixafor	CC chemokine receptor antagonist	ACKR3, **CCR4**, CXCR4, **MMP1**, **PI3**	launched
Ruxolitinib	JAK inhibitor	**JAK1**, **JAK2**, **JAK3**, **TYK2**	launched
Sirolimus	mTOR inhibitor	**CFD1**, FKBP1A, **GPD1**, **MMP1**, **MTOR**, **PI3**, **RPL38**	launched
Tofacitinib	JAK inhibitor	**JAK1**, **JAK2**, **JAK3**	launched
Trofinetide	cytokine production inhibitor	**IFNG**, **IL6**, **TNFA**	2
Upadacitinib	JAK inhibitor	**JAK1**, **JAK2**, **JAK3**	launched
WHI-P154	JAK inhibitor	**EGFR**, **JAK1**, **JAK2**, **JAK3**	preclinical
XL019	JAK inhibitor	**JAK1**, **JAK2**, **JAK3**	1
* **Drugs with known molecular profile registered ** or off-label administered in HS** *			
Acitretin	retinoid receptor agonist	**KRT16**, **PI3**, RARA, RARB, RARG, RBP1, RXRA, RXRB, RXRG, **STAT3**	launched
Adalimumab **	TNF-α inhibitor	**TNF**	launched
Anakinra	IL-1 receptor antagonist	**IL1R1**	launched
Avacopan	C5α receptor antagonist	**C5AR1**	2
Bimekizumab	IL-17A/F inhibitor	**IL17A**, **IL17F**	3
Brodalumab	IL-17 receptor inhibitor	**IL17R**, **KRT6A**, **S100A7A**, **S100A8**, **S100A9**	launched
Clindamycin	Protein synthesis inhibitor		launched
Cyproterone acetate	AR antagonist	ADORA1, **AR**	launched
Doxycycline	bacterial 30S ribosomal subunit inhibitor, metalloproteinase inhibitor	**MMP1**, **MMP8**, **PI3**	launched
Etanercept	TNF-α receptor antagonist	**TNFRSF1A**	launched
Golimumab	TNF inhibitor	**TNF**	launched
INCB 54707	JAK1 inhibitor	**JAK1**	2
Infliximab	TNF inhibitor	**IL6**, **TNF**	launched
Metformin	insulin sensitizer	ACACB, **PRKAB1**	launched
Rifampicin	RNA polymerase inhibitor	NR1I2, SLCO1A2, SLCO1B1, SLCO1B3	launched
Secukinumab	IL-17A inhibitor	**IL17A**	3
Spesolimab	IL-36R antagonist	**IL36RN**	2
Ustekinumab	IL12/IL23 inhibitor	**FSH**, **HCG**, **LH**, **LTA4H**	Launched
Vilobelimab	C5α inhibitor	**C5**	2

* The differentially regulated genes in HS are presented with bold letters.

## Data Availability

Data sets related to this article are hosted at the Gene Expression Omnibus (https://www.ncbi.nlm.nih.gov/geo/ (accessed on 21 November 2021)) data repositories GSE72702, GSE79150, GSE128637, GSE137141, GSE144801, GSE148027, GSE154773, GSE154775, GSE155176, GSE155850 and GSE175990.

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
