# Peer review of "Hidradenitis Suppurativa and Comorbid Disorder Biomarkers, Druggable Genes, New Drugs and Drug Repurposing—A Molecular Meta-Analysis"

_pharmaceutics, 2021, doi:10.3390/pharmaceutics14010044_

Round 1
Reviewer 1 Report
A very well-performed systematic review about the molecular basis of HS and possible new drugs and drug repositioning. I found it really useful, also for clinicians, especially the part of drug repositioning ; there are some minor issues that need to be addressed:
- line 123, the additional records are taken into consideration were six instead of three, as reported in figure 1...please correct.
page 1 line 38-42...this paragraph needs a reference, such as: doi: 10.3390/ijms21228436.
in the subchapter: "Study drugs and drug repurposing for HS" anti interleukin 17 are not mentioned.....being probably the future most promising treatment, they should be mentioned.
Author Response
1st reviewer
Comments and Suggestions for Authors
A very well-performed systematic review about the molecular basis of HS and possible new drugs and drug repositioning. I found it really useful, also for clinicians, especially the part of drug repositioning;
We cordially thank the reviewer for his comments.
there are some minor issues that need to be addressed:
- line 123, the additional records are taken into consideration were six instead of three, as reported in figure 1...please correct.
The mistake was corrected.
page 1 line 38-42...this paragraph needs a reference, such as: doi: 10.3390/ijms21228436.
The reference mentioned by the reviewer was added.
in the subchapter: "Study drugs and drug repurposing for HS" anti interleukin 17 are not mentioned.....being probably the future most promising treatment, they should be mentioned.
Interleukin-17 was added to the promising compounds, as suggested by the reviewer.
Reviewer 2 Report
This meta-analysis review looks into the biomarkers, druggable genes and drug repurposing etc. for Hidradenitis Suppurativa. This is a relative new approach to look into the topics. The article is well written. A minor comment is that how the druggable genes are defined. The authors may provide some explanation to it.
When the abbreviation “HS” is first mentioned in the abstract, it should be explained.
Author Response
2nd reviewer
Comments and Suggestions for Authors
This meta-analysis review looks into the biomarkers, druggable genes and drug repurposing etc. for Hidradenitis Suppurativa. This is a relative new approach to look into the topics. The article is well written.
We cordially thank the reviewer for his comments.
A minor comment is that how the druggable genes are defined. The authors may provide some explanation to it.
The sentence “The interaction level of disease and compound molecular profile patterns defines the probability of therapeutic activity of a certain drug.” was added in lines 56-57.
When the abbreviation “HS” is first mentioned in the abstract, it should be explained.
We have added the term “hidradenitis suppurativa/acne inversa” before the first time of use of the abbreviation “HS” in the abstract.
Reviewer 3 Report
The Authors report about pathogenesis, future treatment and biomarkers in Hidradenitis suppurativa/acne inversa. The work is quite compelling from idea to research. Accept the manuscript after the following minor revision:
1) Each acronym should be explicitated when used for the first time in the text, also in the abstract.
2) About computational drug repurpsing approaches, other methods should be cited such as Di Micco et al. Eur. J. Med. Chem. 2018, 152, 253-263.
3) DEGs refers to genes, but at line 133 is also referred to proteins. Please, clarify.
4) If it is possible simplify the connectivity map
Author Response
3rd reviewer
Comments and Suggestions for Authors
The Authors report about pathogenesis, future treatment and biomarkers in Hidradenitis suppurativa/acne inversa. The work is quite compelling from idea to research.
We cordially thank the reviewer for his comments.
Accept the manuscript after the following minor revision:
1) Each acronym should be explicitated when used for the first time in the text, also in the abstract.
We have added the term “hidradenitis suppurativa/acne inversa” before the first time of use of the abbreviation “HS” in the abstract.
2) About computational drug repurpsing approaches, other methods should be cited such as Di Micco et al. Eur. J. Med. Chem. 2018, 152, 253-263.
The revised text “New technology, including inverse virtual screening [13] and computational drug repurposing screening approaches [14], are widely engaged in identifying existing compounds as potential drugs for various diseases.” Is to be seen in lines 55-58 and the proposed reference has been included.
3) DEGs refers to genes, but at line 133 is also referred to proteins. Please, clarify.
To clarify this point the following comment has been added in lines 136-137.
4) If it is possible simplify the connectivity map.
We would like to keep the connectivity map as it is because despite the 2465 interactions it still characterises well both the complete picture and the connected genes.
Reviewer 4 Report
A valuable review exploring pathogenetic mechanisms at the molecular level and identifying molecular markers of suppurative hidradenitis.
Some minor queries will have to be assessed before granting this paper publication:
Figure 1 should be updated using the last available PRISMA flow chart: http://prisma-statement.org/prismastatement/flowdiagram.aspx
There are some typos in the text (for example, line 325 trias instead of trials)
Some more information about current treatments of HS should be added in the introduction
Page 1 line 38-42; this paragraph needs some referrals, such as: doi: 10.3390/ijms21228436.
Thank You
Author Response
4th reviewer
Comments and Suggestions for Authors
A valuable review exploring pathogenetic mechanisms at the molecular level and identifying molecular markers of suppurative hidradenitis.
We cordially thank the reviewer for his comments.
Some minor queries will have to be assessed before granting this paper publication:
Figure 1 should be updated using the last available PRISMA flow chart: http://prisma-statement.org/prismastatement/flowdiagram.aspx
We have substituted the PRISMA flow chart in Figure 1 with the new PRISMA 2020 flow chart in the revised version of the manuscript.
There are some typos in the text (for example, line 325 trias instead of trials)
In line 325 the word “trias” is correct; it means three signs.
Some more information about current treatments of HS should be added in the introduction
The current registered treatment of HS is already discussed in the chapter “Study drugs and drug repurposing for HS” and shown in table 2.
Page 1 line 38-42; this paragraph needs some referrals, such as: doi: 10.3390/ijms21228436.
The requested reference has been added.
Round 2
Reviewer 1 Report
The authors responded to all queries. the paper is in my opinion ready to be published.
Reviewer 2 Report
The latest version is acceptable for publication.
Reviewer 3 Report
The Authors adequately responded to the revisions raised by this Reviewer. The manuscript can be accepted
Reviewer 4 Report
The authors answered to all queries. The paper is eligible to be published.